# Machine Learning Approach for Muscovy Duck (*Cairina moschata*) Semen Quality Assessment

**DOI:** 10.3390/ani13101596

**Published:** 2023-05-10

**Authors:** Desislava Abadjieva, Boyko Georgiev, Vasko Gerzilov, Ilka Tsvetkova, Paulina Taushanova, Krassimira Todorova, Soren Hayrabedyan

**Affiliations:** 1Department of Immunoneuroendocrinology, Institute of Biology and Immunology of Reproduction, Bulgarian Academy of Sciences, Bul. Tzarigradsko Shosse 73, 1113 Sofia, Bulgaria; 2Department of Animal Science, Agricultural University, 12, Mendeleev Str., 4000 Plovdiv, Bulgaria; 3Reproductive OMICS Laboratory, Institute of Biology and Immunology of Reproduction, Bulgarian Academy of Sciences, Bul. Tzarigradsko Shosse 73, 1113 Sofia, Bulgaria

**Keywords:** Muscovy duck, semen, DNA methylation, biochemical parameters

## Abstract

**Simple Summary:**

We applied machine learning techniques to analyse the kinematics and enzyme activities of Muscovy duck sperm and the DNA methylation levels of the sperm cells. We aimed to find a reliable way to evaluate the quality of duck semen for artificial insemination. We defined good quality semen as having high motility and high methylation. We identified some key features that can predict semen quality, such as the amplitude of lateral head displacement, the wobble and the curvilinear velocity of sperm movement, and the levels of lactate dehydrogenase, alkaline phosphatase, and creatine kinase in the semen.

**Abstract:**

This study aimed to develop a comprehensive approach for assessing fresh ejaculate from Muscovy duck (*Cairina moschata*) drakes to fulfil the requirements of artificial insemination in farm practices. The approach combines sperm kinetics (CASA) with non-kinetic parameters, such as vitality, enzyme activities (alkaline phosphatase (AP), creatine kinase (CK), lactate dehydrogenase (LDH), and γ-glutamyl-transferase (GGT)), and total DNA methylation as training features for a set of machine learning (ML) models designed to enhance the predictive capacity of sperm parameters. Samples were classified based on their progressive motility and DNA methylation features, exhibiting significant differences in total and progressive motility, curvilinear velocity (VCL), velocity of the average path (VAP), linear velocity (VSL), amplitude of lateral head displacement (ALH), beat-cross frequency (BCF), and live normal sperm cells in favour of fast motility ones. Additionally, there were significant differences in enzyme activities for AP and CK, with correlations to LDH and GGT levels. Although motility showed no correlation with total DNA methylation, ALH, wobble of the curvilinear trajectory (WOB), and VCL were significantly different in the newly introduced classification for “suggested good quality”, where both motility and methylation were high. The performance differences observed while training various ML classifiers using different feature subsets highlight the importance of DNA methylation for achieving more accurate sample quality classification, even though there is no correlation between motility and DNA methylation. The parameters ALH, VCL, triton extracted LDH, and VAP were top-ranking for “suggested good quality” predictions by the neural network and gradient boosting models. In conclusion, integrating non-kinetic parameters into machine-learning-based sample classification offers a promising approach for selecting kinetically and morphologically superior duck sperm samples that might otherwise be hindered by a predominance of lowly methylated cells.

## 1. Introduction

In animal and bird breeding, the complete potential of kinematic data obtained from standard Computer Assisted Semen Analysis (CASA) systems is not fully harnessed, despite evidence suggesting the usefulness of certain kinematic parameters in evaluating fertility [1]. Although total motility accounts for only 10% of fertility variation, other kinetic parameters, such as curvilinear velocity (VCL), velocity of the average path (VAP), linear velocity (VSL), and amplitude of lateral head displacement (ALH), have been found to correlate with farrowing rates and the total number of offspring in some mammal species [2]. Additionally, these parameters have been shown to influence fertility prediction models for frozen–thawed semen [3].

Because total motility only explained 10% of the variation in fertility, other kinetic parameters, such as curvilinear velocity (VCL), velocity of the average path (VAP), linear velocity (VSL), and amplitude of lateral head displacement (ALH), were correlated with farrowing rate and total number of piglets born [2]. In agreement with this, another study also reported positive associations between in vitro sperm quality parameters such as motility, viability, normal morphology, and plasma membrane integrity, and the reproductive outcomes of fertility, early embryonic mortality, and survival embryo rate after artificial insemination in Muscovy drakes [4]. These parameters also influenced fertility prediction models for frozen–thawed semen [3].

Machine learning (ML) has been shown to improve the predictive capacity of sperm quality in animal breeding [1]. To overcome the limited predictive power of kinematic parameters alone, recent data-driven machine learning models have integrated extended CASA parameters with other factors that are not related to sperm [5]. The ML technique takes training data and then estimates the predictive ability of the training model by comparing it to test data, constantly evolving it for increased predictive power. This strategy has the potential to considerably improve the efficiency of selecting freeze-thawed sperm dosages with sufficient fertilization capacity, but model fidelity is critical in this process [1]. Therefore, the development of such statistical models necessitates the use of meaningful fertility descriptors and, preferably, the consideration of sperm parameters in both fresh and frozen sperm. Using new functional, non-morphological parameters is very important to additionally increase the trained models’ fitness. 

Both sperm motility competence and chromatin integrity were associated with field fertility and were deemed crucial to the development of fertility prediction models, especially in cryopreserved samples [3]. One reason why the progressive motility and other kinetic parameters of semen are not reliable prognostic markers is that spermatozoa with normal appearance, motility, and chromatin integrity may still have poor reproductive outcomes due to epigenetic changes such as DNA methylation. This could be one of the reasons for the highly inconsistent results for sperm morphological quality assessment in assisted reproduction [6]. Sperm cell epigenetic modifications are more complicated than those of somatic cells for two key reasons. Initially, during the early phase of gametogenesis, when germ cells transform into spermatids, epigenetic modification marks, such as DNA methylated 5-methyl-cytosins (5-meC), are first erased and are then re-established at a later stage, resulting in high DNA methylation. During spermatogenesis, spermatids mature into spermatozoa, and most histones are eventually replaced by protaminase, allowing for substantial reorganizations of sperm chromatin structure. These steps are required for silencing the sperm genetic program before fecundation, and DNA methylation levels are lowered only after the zygote is formed [7,8,9]. Improper DNA methylation is implicated as a primary epigenetic mechanism related to sperm abnormalities, and cytosine methylation at the CpG sites in sperm correlates with gene expression changes, lower fertility, and illness predisposition in humans and other mammals [10,11]. 

Studies on methylation patterns in birds are rare, despite evidence that they resemble mammalian somatic methylation patterns; methylation occurs predominantly on cytosine residues within a CpG context and the majority of the genome is methylated, with the exception of cytosine residues within CpG islands in gene promoters [12]. Interestingly, unlike mammals, birds lack the genetic imprinting phenomenon [13]. Motility and morphological metrics are two indicators of sperm quality that have been linked to DNA methylation patterns in humans [14,15,16]. Few studies have investigated the relationship between epigenetic markers, such as DNA methylation, and sperm quality in birds. None of these studies are relevant to ducks [17,18,19].

The DNA methylation pattern of mature, ejaculated sperm Is highly sensitive to a wide range of internal and external stimuli [18,19,20]. Post-collection semen treatment, such as the addition of various extenders and antioxidants, as well as storage time and temperature management, can positively influence sperm cell survival and extend their fertilizing ability [21,22]. Łukaszewicz et al. (2020) reported that both the duration of semen storage and the choice of extender significantly affected sperm morphology (*p* < 0.05; *p* < 0.01) [21]. The primary role of each extender is to supply sperm with energy, mitigate the physical shear and chemical stress induced by freeze–thaw cycles, and provide an optimal medium for temporary sperm survival [18]. Gerzilov and Andreeva (2021) observed enhanced total motility, with over 80% in semen diluted with IMV Canadyl and AU extenders, and over 68% in semen diluted with HIA-1 extenders [23]. Regarding velocity parameters (VCL, VAP, and VSL), a more rapid decline occurred between 3 and 30 h of cold storage, followed by a slower decrease (*p* < 0.05). The proportion of abnormal sperm cells increases with prolonged storage time [23]. Although kinetic and morphological parameters are traditionally used for sperm quality evaluation, they alone are insufficient for a comprehensive assessment [3,24]. Incorporating DNA methylation analysis can provide valuable insights into epigenetic regulation and its potential impact on fertility [9,16,19,25]. By including this additional parameter, a more comprehensive understanding of sperm function and overall reproductive potential can be achieved, ultimately improving the evaluation of sperm quality.

An additional functional criterion encompasses specific seminal biochemical parameters, such as sperm-produced enzymes such as lactate dehydrogenase (LDH) and creatine phosphokinase (CK), which can reflect sperm maturity, mitochondrial activity, and apoptosis pathways [26,27]. LDH activity has also been directly associated with sperm motility [28,29]. Semen contains low levels of the enzymes CK and LDH, which play important roles in energy metabolism, but whose levels can rise in response to cellular damage or death. Patients with teratozoospermia, a condition that affects sperm morphology, have higher levels of CK and LDH in their semen, as well as a higher proportion of spermatozoa with activated caspases. Caspases mediate apoptosis, or programmed cell death, which can be triggered by various factors, such as oxidative stress, DNA damage, and hormonal imbalance. Apoptosis can negatively affect sperm quality and fertility by reducing sperm count, motility, and viability. Thus, CK and LDH can be used to measure the levels of apoptotic caspase activation in infertile patients. Moreover, biochemical and apoptotic markers are strongly linked to the types and rates of sperm abnormalities, especially in the head, mid-piece, and tail regions [26]. The activity of alkaline phosphatase (AL) in frozen/thawed bovine sperm is associated with the sperm viability fertility parameter. Another enzyme, gamma-glutamyltransferase (GGT), is related to sperm motility and the rates of cleavage and blastocyst formation after in vitro fertilisation. These findings indicate that GGT may have a protective role against oxidative stress in sperm and may serve as a reliable indicator of frozen/thawed sperm quality in bovines [30,31].

The aim of this study is to address the limitations of manual sperm screening in animal breeding, particularly in the context of Muscovy duck (*Cairina moschata*) insemination. Manual screening, which involves visual inspection of sperm morphology and motility under a microscope, is subjective, time-consuming, and prone to human error, and it fails to account for the genetic and epigenetic quality of the sperm that can influence the offspring’s health and performance. In order to overcome these challenges, the study seeks to develop a rapid, non-invasive method that evaluates sperm samples using both sperm kinetics (CASA) and non-morphological functional features such as enzyme activities (alkaline phosphatase (AP), creatine kinase (CK), lactate dehydrogenase (LDH), and γ-glutamyl-transferase (GGT)) in seminal plasma and sperm-derived samples. Additionally, the study will employ machine learning models to analyse total DNA methylation, providing a more comprehensive and reliable assessment of sperm quality for insemination purposes.

## 2. Materials and Methods

### 2.1. Ethics Statement

All experimental procedures were performed under permit No. 220 according the rules of the Ordinance No. 136 of 25 September 2018 on the minimum requirements for protection and welfare of experimental animals and site requirements for use, cultivation, and/or their delivery.

### 2.2. Birds and Rearing

The semen was received from six Muscovy drakes (line CF 80 auto sexing, a product of Grimaud Frères Sélection, France) during the first reproductive period (June–July) in the Poultry Division at the Agricultural University of Plovdiv (latitude 42°14′ N, longitude 24°79′ E, and altitude of 164 m above sea level). The males were clinically healthy, kept individually in spatial metal cages with size 0.6 × 0.8 × 0.6 m placed in a semi-open shed situated under natural light. The birds were fed with a pelleted mixture for breeding drakes comprising: 11.5 MJ/kg metabolize energy, 15.7% crude protein, 4.5% crude fibres, 2.1% crude fats, 1.03% calcium, 0.75% phosphorus total, 0.8% L-lysine, and 0.42% DL-methionine + cysteine. Daily ration was 200–230 g per bird. The intake of water was provided ad libitum.

### 2.3. Semen Collection

The ejaculates were collected two times per week individually by placing a female (teaser method) in the male’s cage using an artificial vagina using the method of Tan [32], modified by Gerzilov [33]. Only ejaculates satisfying the following parameters were included in the study: colour–pearly-white; purity–free of any contamination with cloacal products; volume–above 0.3 mL.

A total of 30 individual semen ejaculate samples (no less than 4 and no more than 6 ejaculates per drake) were evaluated for the purposes of the experiment. Each ejaculate sample (or semen sample) with the use of an automatic pipette was diluted at a ratio 1:3 (*v*/*v*, semen: extender) with an AU extender. The AU extender consisted of the following components: 0.40 g D–glucose, 0.80 g D–fructose, 0.80 g sugar, 0.90 g sodium citrate, 0.84 g sodium glutamate, 0.40 g glycol, 0.04 g ethylenediaminetetraacetic acid disodium salt dihydrate, and plus 100 mL double distilled water. The osmolarity was 320 mOsmol/kg and pH 7.00 [23,34,35].

### 2.4. Sperm Analysis

The diluted semen samples were packaged into semen tubes and transported with electric cool box 12 V DC (capacity 25 L; 4–5 °C) from the Poultry division at the Agricultural University of Plovdiv to the Institute of Biology and Immunology of Reproduction at Bulgarian Academy of Sciences, Sofia. The samples were analysed after 6 h storage of diluted semen. 

The assessment of semen quality parameters was carried out with a CASA system [Sperm Class Analyzer (SCA)], version 6.4.0.99 (Microptic, Barcelona, Spain), acquiring frames at 60 Hz. The main components of the SCA system were a Nikon microscope (Eclipse E200) with negative phase contrast optics, a charge-coupled device (CCD) video camera, and a personal computer with a specific software. 

The evaluation of semen quality parameters was conducted using a Sperm Class Analyzer (version 6.4.0.99 CASA system) from Microptic (Barcelona, Spain), comprised of a Nikon Eclipse E200 microscope with negative phase-contrast optics, a Basler Ace charge-coupled device video camera, and a computer running a dedicated SCA Evolution CASA software. Image frames were captured at a rate of 60 Hz, with default CASA detection parameters: sperm concentration (>15 million/mL), total sperm count (100), percentage of motile spermatozoa in the sample (>40%), and spermatozoa classification into slow–medium (50 µm/s), rapid (>100 µm/s), and progressive (straight-line velocity, STR, >100 µm). A 5 µL sperm sample, diluted to a concentration of 50 × 10^6^, was loaded into a pre-warmed, disposable analysis chamber with a 20 µm depth (Leja Products, Nieuw-Vennep, The Netherlands) and analysed on the pre-warmed stage. For each slide, 100 spermatozoa were analysed at 10× magnification. Cells with an average path velocity (VAP) greater than 25 µm/s and a straight-line velocity (VSL) greater than 100 µm/s were classified as progressively motile cells. Semen samples were also analysed for sperm vitality using the BrightVit kit (Microptic, Barcelona, Spain).

### 2.5. Enzyme Assay

To obtain seminal plasma, the ejaculate was centrifuged at 3000 rpm for 10 min. The procedure was repeated with the supernatant of the first centrifugation. The obtained seminal plasma was decanted and processed for next analysis. Pellet, received after first centrifugation was resuspended with saline and centrifuged at 3000 rpm/10 min. The procedure was repeated 3 times. After last centrifugation, pellet was resuspended in distilled water and stored at −20 °C overnight. Next day, after thawing, pellet was sonicated 3 times × 10 sec and centrifuged at 12,000 rpm for 30 min. Received supernatant contains water soluble proteins, extracted from sperm cells. The pellet was resuspended in Triton X100, centrifuged at 12,000 rpm for 30 min, and supernatant was used for analysis of membrane connected proteins in spermatozoa. Each diluted sperm sample was further divided into three equal parts-seminal plasma (sp), water sperm extract (w), and Triton extract (t). Both extracts and seminal plasma were used to analyse the enzyme activity of alkaline phosphatase (AP) [30,31], creatine kinase (CK) [27,36], lactate dehydrogenase (LDH) [29], and γ-glutamyl-transferase (GGT) [24] by biochemical semi-automatic analyser BA-88A (Mindray, China), following Via Campania kits (Italy) manufacturer instructions: for GGT- Tris buffer 100 mM pH 8.25,glycil-glycine 100 mM, L-Glutamyl-4-nitroanilide 4 mM, wavelength 405 nm; for AP-DEA buffer pH 9.8 1 M, MgCl2 0.5 mM, 4-Nitrophenilphosphate 10 mM, wavelength 405 nm; for CK- Imidazole buffer 29 mM pH 6.50, creatine phosphate 30 mM, glucose 20 mM, N-Acetyl-L-cysteine 20 mM, magnesium acetate 10 mM, EDTA 2 mM, ADP 2 mM, NADP 2 mM, AMP 5 mM, Di(adenosine-5) pentaphosphate 12 MikroM, glucose-6-phosphate- dehydrogenase > 3 kU/L, hexokinase > 3 kU/L, wavelength 340 nm; for LDH-phosphate buffer pH7.50 50 mM, sodium pyruvate 0.60 mM, NADH 0.18 mM, wave length 340 nm. Distilled water was used as an assay standard, which was analysed in the system before measuring the samples. This statement appears to have been written intentionally. The enzyme activity was measured in units per litre [U/L] and annotated using prefixes (sp, w, or t) added to each enzyme abbreviation to indicate the specific enzyme activity being measured.

### 2.6. DNA Methylation

DNA methylation is known to play a crucial role in sperm fertility, as well as in embryo vitality and development [9,16,37,38,39]. In each sample, DNA methylation was assessed by determining the percentage of the 5-methylcytosine (5-meC) epigenetic mark in DNA relative to a fully methylated DNA control in fresh Muscovy duck semen samples. This was performed using a 5-meC ELISA kit (Zymo Research, Tustin, CA, USA) according to the manufacturer’s instructions [25]. In brief, total genomic DNA was extracted from the samples using the Quick-DNA Plus kit (Zymo Research, USA) and then subjected to 5-meC detection using a monoclonal anti-5-Methylcytosine antibody, which is sensitive and specific to this base modification.

### 2.7. Machine Learning Sperm Parameters Classification

Sperm samples were characterized using 35 parameters: total DNA methylation – 5-meC [%]; CASA kinetic parameters: sperm concentration [10^6^/mL], progressive motility [%], non-progressive motility [%], Static [%], Rapid [%], Medium [%], Slow [%], VCL, VAP, VSL, STR, LIN, WOB, ALH, BCF, normal [%], head [%], mid piece [%], tail [%], cd [%]; non-kinetic parameters: sperm staining-live [%], dead [%]; sperm enzyme activity: spAP, spLDH, spCK, spGGT, wAP, wLDH, wCK, wGGT, tAP, tLDH, tCK, tGGT. Sperm samples were additionally labelled using three new classification variables for ML model training according to the following criteria: 1. Variable “Quality” denoted commonly acknowledged quality based solely on progressive motility. Label “Quality” assumed value “Good” when progressive motility percentage was above 50%, otherwise it assumed value “Bad”; 2. Variable “High methylation” denoted samples total DNA methylation percentage. Label “High methylation” assumed value “High”, when 5-meC percentage was above 50%, otherwise it assumed value “Low”; 3. Variable “Suggested Good Quality” denoted those samples that had commonly accepted good progressive motility and sufficiently high total DNA methylation. Variable “Suggested Good Quality” assumed value of “Good expectation” in case both conditions were met: “Quality” was equal to “Good” and “High methylation” was equal to “High”, otherwise it assumed value “Bad expectation”.

### 2.8. Training of Machine Learning Models

Orange Data Mining software [40] was used to build several classifiers on the samples investigated, including Random Forest, Adaptive Boosting (Ad Boost), Gradient Boosting, a multi-layer perceptron algorithm with backpropagation (Neural network), Support Vector Machine map (SVM), kNN (k-Nearest Neighbours), and Naïve Bayes, using either the classification variable “Quality”, “High Methylation”, or “Suggested Good Quality” for supervised training of the models with the labels described above. Orange uses visual programming and Python script to provide a controlled user interface for entering data and performing machine learning and data mining analysis. Most of the ML models are based on their implementation in the open access Python scikit-learn machine learning programming library. Neural network was a multi-layer perceptron algorithm with backpropagation (implemented in scikit-learn), with neurons per hidden layer: 200, Activation function for the hidden layer: Logistic (logistic sigmoid function), Solver for weight optimization: Adam (stochastic gradient-based optimizer), and Max training iterations: 200. Other models were used with their default parameters set in Orange widgets and scikit-learn library.

In summary, the training process involves feeding a set of sperm quality parameters into machine learning models, which then learn to recognize patterns and make predictions accordingly. The classifiers were trained to evaluate parameters for semen quality obtained using CASA, semen enzyme activity, and global DNA methylation. Training was performed on the entire dataset or by omitting variables related to CASA or the 5-meC percentage when these parameters were linearly correlated with the parameter used for labelling. The actual model training consists of providing a training set of data (a portion of the entire dataset of semen samples, each characterized by CASA, enzyme, and DNA methylation parameters) and supplying values for the defined labels-variable (classification label) “Quality”, variable “High methylation”, and variable “Suggested Good Quality”. Once the models were trained, they were able to produce predictions of the new label values (variables “Quality”, “High methylation”, “Suggested Good Quality”) based on the input parameters. In this case, the models output classifications for sperm samples from other previously “*unseen*” datasets into categories such as “Good” or “Bad” quality, “High” or “Low” methylation, and “Good expectation” or “Bad expectation” for suggested good quality. Each model’s predictive ability is tested against an “unseen” dataset, which is a portion of the entire dataset that was not used for its training. The method used is called “*k-fold cross-validation*” and it can evaluate the performance of a machine learning model on unseen data. It involves dividing the dataset into k smaller sets (or folds) and using one of them as the test set and the rest as the training set. This process was repeated 5 times (k = 5), each time using a different fold as the test set. The Test & Score widget in Orange implements cross-validation and reports the average performance of the model across all subsets. 

### 2.9. Performance Assessment of the Models (Validation)

The performance of a model can be assessed by different metrics, such as accuracy, precision, recall, F1-score, or AUC, depending on the type and objective of the model. The average performance across all k runs is reported. The model performance is defined by a complex set of estimated metrics and is described in detail below. Verification is the process of verifying whether the model satisfies the problem requirements and specifications. It can be performed by comparing the model predictions with the actual outcomes on the test set and calculating metrics such as a confusion matrix. A confusion matrix is a table that summarizes how well a machine learning model performs on a set of test data with known true values. It is used to measure accuracy, recall, precision, specificity, and other metrics for classification models with two or more classes. A confusion matrix shows the number of *true positives* (TP), true negatives (TN), false positives (FP), and false negatives (FN) that the model produces on the test data: TP-the model predicts the positive class correctly, TN-the model predicts the negative class correctly, FP-the model predicts the positive class incorrectly, FN-the model predicts the negative class incorrectly.

A confusion matrix helps to compare ML models and evaluate their performance by computing the metrics accuracy, recall, precision, and F1-score from the values in the table; they were used to rank the learner models. In an uneven dataset, accuracy (the percentage of true predictions) may be skewed toward the more numerous majority class (calculated as (TP + TN)/(TP + TN + FP + FN)). Classification inside the learning model was made objective by calculating recall, precision, and F1-score. The recall rate is the average percentage of properly categorized sperm samples across all quality categories (calculated as TP/(TP + FN)). Precision measures how well a system can make favourable predictions (calculated as TP/(TP + FP)). Because there is a trade-off between precision and recall, F1-score was chosen as the major parameter for assessment. The F1-score reflects the total “proficiency” of the classifier in a single assessment metric, as it is the harmonic mean of precision and recall (2 × precision × recall/(precision + recall)). AUC metric measures the area under the receiver operating characteristic curve. It is a measure of how well a model can separate different classes. It ranges from 0 to 1, where 1 means perfect separation and 0.5 means random prediction. All model-based learners, produced via the 5-fold cross-validation procedure, were used to compute performance measures.

### 2.10. Model Predictions Utilization

Orange software allows all trained model weights and parameters to be saved in separate file and later to be reloaded for prediction use. Then new datasets and predictive models (preliminary saved from the trained model) could be fed to the Predictions widget, allowing for the estimation of (predicting) the classifier label variable values of the new dataset. This allows the model to be continuously extended and used by other users.

### 2.11. Statistical Analysis

Data were subjected to one-way analysis of variance (ANOVA) followed by t-test to determine the level of significance among mean values. The results are presented as mean ± standard error of mean. The significant differences among mean values were determined by Duncan’s multiple range test at significance level *p* < 0.05. Spearman’s rho correlation test was used for analysis of correlations among the different parameters.

## 3. Results

### 3.1. CASA Parameters Differ between Groups with High and Low Progressive Motility Labelled as “Good” vs. “Bad” Quality

The fertilizing capacity of in vitro preserved sperm can be prolonged by diluting avian sperm with a suitable extender. The fundamental sperm parameters and DNA methylation of Muscovy duck (*Cairina moschata*) drake semen were studied in this work (Table 1). The analyses of the data showed significant differences for total and progressive motility, rapid spermatozoa, VCL, VAP, VSL, ALH (*p* < 0.001), and BCF and live normal sperm cells (*p* < 0.05) in favour of the “Good” quality semen samples over those of “Bad” quality (based on label “Quality” criteria). In contrast, significantly more non-progressive motility, slow, static, and abnormal spermatozoa (*p* < 0.001) were reported in the bad-quality sperm samples compared to the good-quality ones (Table 1).

Semen sample enzyme activity was significantly different for enzymes AP (*p* < 0.05) and CK (*p* < 0.01), among the groups classified by the label “Quality” (based on progressive motility). It was found that the LDH and CK enzyme activities were significantly different in the sperm media (*p* < 0.05 and *p* < 0.05, respectively). The types of sperm enzyme extracts also had an effect on the acquired enzyme activities. We found significant difference in enzyme activities of AP (*p* < 0.01), CK (*p* < 0.01), and GGT (*p* < 0.05) among different sperm extract types, but only in sperm samples classified as “Good” by parameter “Quality”. The interaction “Semen Quality × Medium” also significant affected enzyme activities of AP, CK, and GGT. Overall, sperm samples classified as “Good” had higher LDH and GGT levels, whereas sperm samples classified as “Bad” had lower AP and CK levels, based on “Quality” labelling (Table 2).

There was only a weak direct correlation (r < 0.6) between the values of the CASA parameter progressive motility percentage, which we used to classify samples as “Good” under the “Quality” label, and the enzyme activities of AP, CK, and GGT (Figure 1). All correlation coefficients (rI) were significant (*p* < 0.001), with the exception of r_Good_ (progressive motility percentage, spAP), r_Bad_ (progressive motility percentage, spAP), and r_Bad_ (progressive motility percentage, wCK), which were estimated within class samples. The total DNA methylation values did not correlate with semen concentration, total motility, or vitality (Appendix A), although there were significant correlations between these later fundamental semen parameters and the enzyme activity (Table 2). The data heterogeneity suggested that total DNA methylation is a phenomenon that does not directly affect semen total motility, nor vitality.

### 3.2. Methylation Level Could Be Used to Potentially Stratify the Sperm Characterization Parameters

In order to address the motility and other kinetic parameters’ heterogeneity, as well as to support the better stratification, we used the total DNA methylation per sample as an additional labelling criterion. The presumption was that despite the potential excellent motility and good morphology, sperm with very low methylation could not participate in viable embryo development. Both gametes are required to have high methylation level and silencing of almost all their genetic program [7,8,9,10,11]. Thus, we added two new labels: “High methylation” and “Suggested Good Quality”, with the latter assuming a “Good expectation” (or “Good”) value when “Quality” was “Good” and “High methylation” was “High” (if the 5-meC percentage value was greater than 50%). There was no significant difference between the “High” and “Low” methylated samples (label “High methylation”) when comparing progressive motility, hence the “Quality” label across the total DNA methylation status, but there was significantly different distribution of the progressive motility values between “Suggested Good Quality” classified data, with highest progressive motility attributed to the “Good expectation” samples (Figure 2, Appendix A). When we applied “Suggested Good Quality” labelling to the dataset, we found that sperm samples were significantly different by their values for CASA parameters ALH, VCL, and WOB, with “Good expectation” samples having highest ALH and VCL values, and lowest WOB values (Figure 3). We similarly analysed all other CASA and enzyme parameters for significance towards “Suggested Good Quality”, but there was no difference between “Good expectations” vs. “Bad expectation” values (Appendix A).

### 3.3. ML Classifications Suggest Which CASA and Enzyme Parameters Could Potentially Be Used Together with DNA Methylation Status for Field Preliminary Analysis of Samples

In order to classify the heterogeneous dataset in an unbiased manner, we performed a cross-validation procedure of training learning algorithms using five folds (equal parts) of the dataset, each time using a separate fold as the test dataset and the remaining folds as the training dataset. We initially examined whether the classifier “Quality” could be reliably predicted using a subset of the sperm characterization’s 18 parameters (spAP, spLDH, spCK, spGGT, wAP, wLDH, wCK, wGGT, tAP, tLDH, tCK, tGGT, ALH, WOB, 5-meC perc, VCL, VSL, VAP). We purposefully excluded additional characteristics that may be linearly related to increasing motility. We found four learning algorithms to have high accuracy, recall, precision, and F1 score–Gradient Boosting model (scikit-learn), Gradient Boosting Model Random Forest (xgboost), Random Forest, and Neural Network (Multi-layered Perception algorithm implemented in scikit-learn, MLP-NN), (Figure 4a,b, Appendix A). Assuming the recall reflects the true positive rate, while F1-score represents the overall ability of the classifier, we found Gradient Boosting and Extreme Gradient Boosting RF (Random Forest) to outperform Random Forest and Multi-layered Perception Backpropagation Neural Network (MLP-NN) to predict the class “Quality” values using the 18 sperm parameters listed above. Gradient Boosting algorithms had 93% for both F1-score and recall, while Random Forest and Neural Network had only 89.7% and 90% F1-score and Recall, respectively. Confusion matrix for Extreme Gradient Boosting Random Forest is presented, showing model performance in “Quality” classes “Good” vs. “Bad”. Of all actual “Bad” labelled samples, 4.3% were classified as False “Good”, while another 14.3% of “Bad” classified samples were labelled “Good” (Figure 4c). From all features, VCL, ALH, tGGT, VAP, and tCK were shown as the top five most influential on the classifier (Appendix A). The correlation between VCL and ALH parameters was very strong (r = 0.98) when only correctly classified samples by Gradient Boosting Random Forest learner were used (Figure 4d). Receiver operating characteristic (ROC) curves, Calibration curve, and Performance curve of the four compared ML models are shown (Figure 4e–g). The ROC curve illustrates the relation between the model’s true positive rate and false positive rate, with an ideal curve touching the upper left corner of the plot. The ROC curve illustrates the relation between the model’s true positive rate and false positive rate, with an ideal curve touching the upper left corner of the plot. ROC curves of Random Forest and Neural Network (NN) performed very well, but only until the recall reached 70%, while Gradient Boosting learners showed an initial small false positive rate that was later compensated by high recall (Figure 4e). In contrast, Calibration Plot plots class probabilities against those predicted by the classifiers, showing the prediction accuracy of class probabilities in a plot. At the bottom of the graph, the points to the left are those which are (correctly) assigned a low probability of the target class, and those to the right are incorrectly assigned high probabilities. At the top of the graph, the instances to the right are correctly assigned high probabilities, and vice versa. This graph indicates whether a classifier is too optimistic (provides primarily positive results) or pessimistic (gives predominantly negative results). Random Forest and SLP-NN are overly pessimistic, while Extreme Gradient Boosting Random Forest is overly optimistic, leaving Gradient Boosting with best prediction accuracy of classes within “Quality” classification feature, although both Boosting algorithms have identical recall and F1 values (Figure 4f). The Performance Lift Curve compares the performance of a selected classifier to that of a random classifier and is frequently used for population segmentation. The *X*-axis represents the population (P-rate), while the *Y*-axis represents the true positive (TP-rate). The lift on each point is calculated using the positive real labels of the proportion of the sample of the data up to each point, and the total number of positive real labels of our data. The higher the upper left point, the better the model’s performance because there are a large number of true positive labels in a subset of the investigated data, which has a very high possibility of being positive. We observe that Extreme Gradient Boosting here also performs worse than the Gradient Boosting model (Figure 4g).

In summary, Gradient Boosting performed better in predicting the “Quality” class, with VCL, ALH, tGGT, VAP, and tCK being most influential on the classifier. It had the highest true positive rate (recall 93%), as well as overall predictive ability (F1 93%).

Similar to the approach described so far, we analysed whether the classifier “High Methylation” could be reliably predicted using a subset of the sperm characterization’s 17 characteristics (spAP, spLDH, spCK, spGGT, wAP, wLDH, wCK, wGGT, tAP, tLDH, tCK, tGGT, ALH, WOB, VCL, VSL, VAP), where progressive motility percentage and 5-meC methylation percentages were omitted from the training and following 5-fold cross-validation. Following features ranked as most influential for the classifier: tLDH, wLDH, spLDH, ALH, tGGT. Unfortunately, there was not a single learner algorithm to produce a model with accuracy, precision, and recall above 80% (Appendix A), hence we concluded that this parameter set and amount of data are not sufficient to train total methylation level detection.

We finally examined whether the novel classifier variable “Suggested Good Quality” could be reliably predicted using the first subset of the sperm characterization’s 18 characteristics (spAP, spLDH, spCK, spGGT, wAP, wLDH, wCK, wGGT, tAP, tLDH, tCK, tGGT, ALH, WOB, VCL, VSL, VAP). Both Progressive motility percentage and 5-meC percentage total methylation features were deliberately omitted as directly involved in the classifier variable value determination. We found three learning algorithms to have high accuracy, recall, precision, and F1 score–Neural Network (Multi-layer Perceptron scikit-learn algorithm, MLP-NN), Ad Boost (Orange), and Gradient Boosting (scikit-learn) (Figure 5a–c), (Appendix A). MLP-NN outperformed the other two top performers with 100% recall (true positive rate) and F1-score (overall classifier ability), while Ad Boost and Gradient Boosting (scikit-learn) both scored 97.55% recall/F1-score. Generally, all three learning algorithm models trained using above 17 parameters ranked very high in predicting the “Suggested Good Quality” classes “Good expression” and “Bad expectation” (Figure 5b). It is to be noted that the variable total methylation percentage (5-meC_perc) was not part of the training dataset, or part of the validation dataset in the several fold cross-validation procedures. All features ranked by model influence, ALH, VCL, tLDH, VAP, and wLDH were shown as top five scoring (Appendix A). The correlation between the VCL and ALH parameters was very strong (r = 0.98) when only correctly classified samples by MLP-NN learner were used (Figure 5d). We also found that for correctly predicted data from the dataset, when pre-classified using newly introduced criteria for methylation and suggested good quality, there were several strong correlations between different enzyme activities (Appendix A).

ROC curves of MLP-NN and Gradient Boosting showed that models performed very well (Figure 5e). The Calibration Plot showing the prediction accuracy of class probabilities identified MLP-NN to have an optimal classification of the samples in the classes defined for the “Suggested Good Quality” classification variable. The Gradient Boosting algorithm (scikit-learn) was overly optimistic, while Ad Boost was only slightly pessimistic in its predictions, despite the identical recall and F1 values the two later learners had (Figure 5f). Performance Lift Curve measuring the performance of the three classifiers against a random classifier showed that the Ad Boost and Gradient Boosting models (scikit-learn) kept high performance even at data with low probability to be positive. All three had equal top performance (Figure 5g). Summarizing, the MLP-NN and Ad Boost were topmost performers overall, considering all three performance evaluations.

In summary, three learning algorithms displayed high accuracy, recall, precision, and F1 score: MLP-NN, Ad Boost, and Gradient Boosting. MLP-NN outperformed the others with 100% recall and F1-score, while Ad Boost and Gradient Boosting both scored 97.55% in these metrics. The top five influential features were ALH, VCL, tLDH, VAP, and wLDH. The ROC curves for MLP-NN and Gradient Boosting indicated excellent model performance. The Calibration Plot revealed that MLP-NN had optimal classification for the “Suggested Good Quality” variable, while Gradient Boosting was overly optimistic and Ad Boost was slightly pessimistic. The Performance Lift Curve showed that Ad Boost and Gradient Boosting maintained high performance even with low probability data, and all three models had equal top performance. Overall, MLP-NN and Ad Boost were the top performers across all three performance evaluations.

Finally, we carried out several tests to determine if we have avoided overfitting of the trained learner algorithms because the dataset was relatively small in size. We found that by scrambling the dataset classification parameters (labels), we obtained significantly worse accuracy and other performance parameters rates, suggesting that our model is not over fitted (Appendix A). The Neural Network Calibration curve also showed no signs of overly optimistic, nor overly pessimistic, prognosis.

The Classification Model explanation plot (Figure 6) showed positive high ALH, VCL, and VAP values and low WOB value to increase the probability “Good expectation” class to be predicted. Similarly, positive LDH and AP, and negative CK values could also potentially increase the probability of such prognosis. Correctly predicted for “Suggested Good Quality” samples had significant correlation between their ALH values and numerous CASA parameters (Appendix A).

## 4. Discussion

The aim of this study was to develop a comprehensive method for evaluating fresh ejaculate from Muscovy duck (*Cairina moschata*) drakes to fulfil the needs of artificial insemination practices in agriculture. Currently, sperm samples are evaluated based on their kinetic parameters, as determined by the Computer-Assisted Sperm Analysis system, with a particular focus on progressive motility percentages exceeding 50%. Additionally, some laboratory procedures employ vital dye staining and manual microscopic examination of sperm morphology at high magnification [1,5]. It should be noted that most of the studies on “quality of the sperm” are performed with humans and in livestock breeding, while studies performed with birds are relatively limited to date. We added a classification feature for “good quality sperm” based on more than 50% progressive motility as a starting criterion. Although in this study we found a good correlation between progressive motility and CASA parameters such as ALH, and moderate to high correlation between ALH and other CASA parameters, there was poor correlation between the spermatozoa head movement defining ALH or progressive motility from one side, and vitality staining, or other spermatozoa compartments’ motility parameters. This led us to introduce another set of non-kinetic parameters including four enzymes–AP, CK, LDH, and GGT. Results showed significant differences for total and progressive motility, rapid spermatozoa, VCL, VAP, VSL, ALH, and BCF and live normal sperm cells. The semen samples had a significant effect on AP, LDL, CK, and GGT levels. In summary, the high degree of heterogeneity observed within the limited dataset precluded the establishment of definitive cut-off criteria based on a single parameter. Consequently, a panel of parameters would be better suited for describing the potential of the ejaculate for further use. Because the morphology and motility of an intact sperm are only part of the spermatozoa potential for producing a viable embryo, we introduced another, epigenetics-based parameter–total DNA methylation, based on the percentage of 5-meC methylation vs. total DNA methylated control. This is the first time such study has been performed on ducks, and there is a very limited number of studies performed with other birds. Environmental factors affect epigenetic markers throughout development. DNA methylation is a key epigenetic process in developmental biology that plays crucial roles in sex chromosomal dosage compensation and genome integrity [39]. Identifying these characteristics is crucial to improving cattle and poultry phenotypes. Sperm DNA methylation undergoes reprogramming events that make it distinct from other somatic cells and tissues, as histones are replaced with protamine, and sperm becomes hyper methylated, compared to somatic cells [17,18,19]. Both gametes have high DNA methylation before forming a zygote. During the early zygote stages of development, a complete DNA demethylation event takes place. Subsequently, as the blastocyst cells continue to develop, they reacquire a certain level of DNA methylation [38,41]. This phenomenon suggests that low DNA methylated sperm is highly unlikely to produce a viable zygote and normal embryo development process hereof, which is already confirmed by studies in human and mammalian species [37]. Therefore, we introduced two more classification features (labels) that describe above the 50% level of total DNA methylation, and one that combined both spermatozoa features–high progressive motility with high methylation status. Interestingly, sperm motility was significantly increased in the sperm samples with high methylation, allowing us to define a group of samples that could be termed “Good expectation” (feature “Suggested Good Quality”). Additionally, DNA methylation may influence male fertility and sperm quality through the link between enzyme activity and important biochemical components. AP has a function in maintaining the quiescence of spermatozoa until they are ejaculated and in modifying the acquisition of fertilizing capacity [31]. Semen has elevated levels of creatine (Cr) and higher CK activity. Low semen Cr is related with diminished sperm motility, but high CK activity is associated with worse sperm quality. The results indicate that Cr is essential for sperm metabolism and that supplementation with Cr may be beneficial for individuals with low-quality sperm [36]. GGT could play an important role in the protection of sperm against oxidative stress and could be considered a reliable marker to assess sperm quality [30]. Lactate Dehydrogenase-C4 (LDH-C4) is also required for sperm function. Absence of the enzyme impairs ATP synthesis in sperm, sperm function, and fertility [42,43].

We used a machine learning approach in this study to elaborate on the more complex interactions between the DNA methylated state, kinetic CASA, and non-kinetic enzyme parameters. Machine learning has become the predominant approach for assisting experimental research. Supervised ML has also been effectively used in studies of sperm quality [44,45]. However, in biomedicine and animal breeding, aside from genomics data, the sizes of the datasets are limited. In this regard, we used several different learning algorithms to identify studied relations, but also to elaborate on a long-term protocol for sperm quality analysis. We found that SLP neural network and gradient boosting algorithms outperformed the more traditional options such as Naïve Bayes, Random Forest, SVM, and kNN. This could be partially due to the introduction of several condition-based labels and reducing the feature set number by removing the features that directly participated in the definition of the new labels (classification features). We performed several steps to reduce the risk of overfitting due to the small dataset [46], including weak supervision [47,48], where noisy training labels are generated by labelling functions that each provide a label for some subset of the data, and collectively generate a large but potentially overlapping set of training labels. In our case, “Quality”, “High Methylation”, and “Suggested Good Quality” served that purpose. This could partially explain the high accuracy performance of SLP-NN and gradient boosting algorithms. Studies on the impact of small dataset sizes suggest that the overall performance of classifiers depends on the extent to which a dataset represents the original distribution rather than its size. A robust machine learning model to dataset size reduction does not necessary imply that it provides the best performance compared to other models. This was evident by the observation that Ad Boost and Naïve Bayes models were the most robust models to dataset size reduction in a study, but they had the least average accuracy on the small datasets, while Random Forest and Neural Network showed a relatively similar response to the decrease of dataset size but show significant performance degradation [46], similar to our own data analysis. We also had SLP-NN and Gradient Boosting algorithms having best accuracy, which degraded when scrambled classification features were fed to them. Different training sets for classifying properly high methylation and good quality with high methylation suggested that high methylation is a very important parameter for the synthetic scenario, while some head and tail kinetic parameters (ALH, WOB) and enzyme activities (LDH, GGT) were more important for the entire sample set, but also performed very well in the “Suggested Good Quality” classification feature. Changing the training set with omitting the total DNA methylation revealed that this feature is important for proper classification not only of high-methylation-featured samples, but also of the double positive criterion “Suggested Good Quality” (weak supervised training) datasets, suggesting an internal dependence on the methylation that was not directly derivable from the other parameters.

When all data were fed, or some parameters were omitted in purpose, the SLP-NN classifier was the only classifier that had not a single error prediction value. Cross-validation revealed that when several classifiers were trained using only parameters that were not linearly dependent on the percentage of sperm progressive mobility on which the training label for quality is based, the ranking of the classifiers shifted in favour of Random Forest and others. The AUC and precision of Neural network and Naïve Bayes methods declined. Then, we omitted the total DNA methylation data and discovered that the error rate in true positive values decreased when classifiers were evaluated, despite the fact that sample total DNA methylation had no correlation with the “good” quality parameter or others. This suggests that DNA methylation plays some role in sperm quality determination, but it is not the only parameter involved. One of the main advantages of SLP-NN is that it is a data-driven self-adaptive method, in that it is adjustable to the data without the need for explicit specification of the underlying model. Another feature of SLP-NN is that it represents a nonlinear, model-free method [49,50,51,52]. All of these features demonstrates the usefulness of neural networks to the evaluation of individual sperm samples. When a minimal set of parameters was fed to the ML classifiers used, among the top five parameters with influence on the classifier predictability, ALH, GGT, and CK were selected. Although measures were taken to cope with the dataset size, increasing the dataset size would significantly increase the classifier robustness. Additionally, more datasets could be used to test the classifier, and DNA sequencing data or other molecular markers could be introduced. Unfortunately, there is no single dataset on morphology or kinetics that is available so far for ducks to test the trained classifiers. Interspecies differences in these parameters are significant [1,3], preventing other species data from being freely used. 

The machine learning approach presented in this study offers several advantages, including the potential for expanding the training dataset and enabling others to utilize the pre-trained models in Orange with their own data. This data can be composed of all training parameters or a subset of them, and will still yield relatively accurate predictions. This approach allows for practical applications because it eliminates the need for each user to train their own data, thereby promoting accessibility and efficiency.

## 5. Conclusions

In conclusion, incorporating non-kinetic parameters and total DNA methylation into machine-learning-based classification presents a promising strategy for selecting kinetically and morphologically superior duck sperm samples that might otherwise be hindered by a predominance of lowly methylated spermatozoa cells. The use of machine learning for sample classification and feature evaluation further underscores the significance of DNA methylation as a crucial parameter in assessing sperm quality.

## Figures and Tables

**Figure 1 animals-13-01596-f001:**
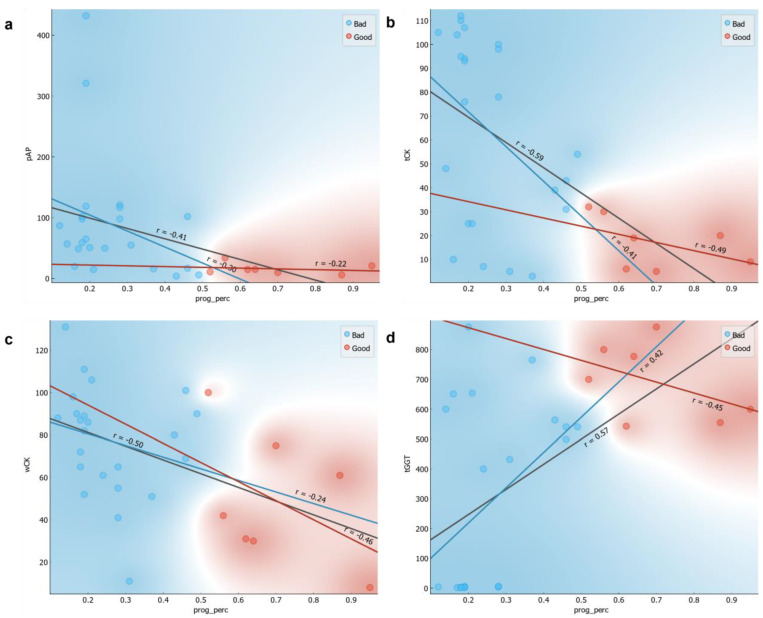
Correlation between the CASA parameter progressive motility (prog_perc) [%] and enzyme activity spAP, tCK, wCK, tGGT. All correlations (r) are significant (*p* < 0.001), except following r_Good_ (prog_perc, spAP), r_Bad_(prog_perc, spAP), r_Bad_(prog_perc, wCK), which were estimated within class samples. (**a**) Correlation between progressive motility and spAP; (**b**) Correlation between progressive motility and tCK; (**c**) Correlation between progressive motility and wCK; (**d**) Correlation between progressive motility and tGGT.

**Figure 2 animals-13-01596-f002:**
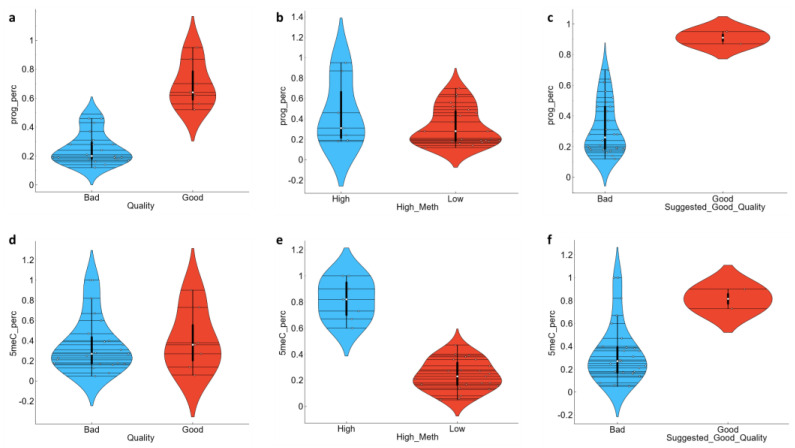
Violin plot showing the distribution of Progressive Motility and Total 5-meC modifications in semen ejaculate samples grouped according to the classification features “Quality” (**a**,**d**), “High Methylation”(**b**,**e**), and “Suggested Good Quality”(**c**,**f**). Both parameters are measured in percentages and are represented by values in the range 0 to 1. The values of the classification features are presented on the *x*-axis, while the distribution is on the *y*-axis.

**Figure 3 animals-13-01596-f003:**
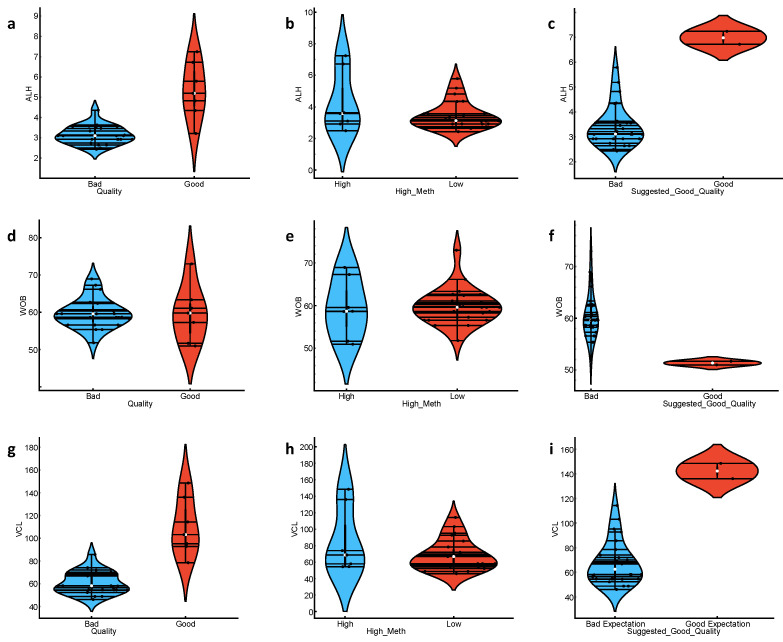
Violin plots of CASA parameters ALH, WOB, and VCL distribution across classification features “Quality” (**a**,**d**,**g**), “High Methylation”(**b**,**e**,**h**), and “Suggested Good Quality”(**c**,**f**,**i**). The values of the classification features are presented on the *x*-axis, while the distribution is on the *y*-axis.

**Figure 4 animals-13-01596-f004:**
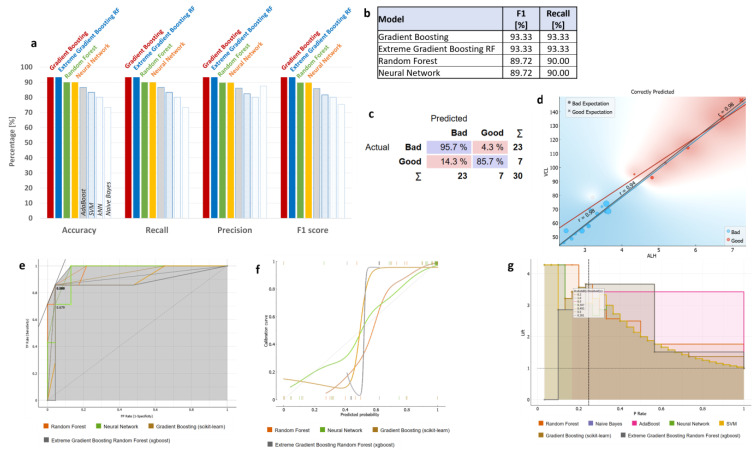
Classification power of the learning algorithms used to classify the sperm dataset based on classifying label “Quality”. Predictions were made on the dataset in a 5-fold cross-validation process, where the total dataset was split into 5 folds and each was once used as a test set, while the remaining ones were used as a validation set. Each cross-validation repetition used a different seed. Metrics presented are Orange cross-validation algorithm performance parameters: (**a**) The percentage accuracy, recall, precision, and F1-score of the different ML algorithm models are compared with each other; (**b**) F1-score and recall from the Gradient Boosting (scikit-learn), Extreme Gradient Boosting Random Forest (xgboost), Orange Random Forest, and Neural Network (Multi-layer Perceptron *scikit-learn* algorithm) learning models considerably outperformed the Ad Boost, SVM, kNN, and Naïve Bayes algorithms; (**c**) Confusion matrix of the Extreme Gradient Boosting Random Forest model showing actual probabilities of correct and incorrect classification for progressive-motility-based “Quality” class, where blue shading shows correctly classified predictions; (**d**) Example correlation plot of parameters selected based on the best classifier, with the strongest correlation (*p* < 0.001 for r and r_Good_, r_Bad_); (**e**) Receiver operating characteristic (ROC) curves; (**f**) Calibration curves; (**g**) Performance curve of the compared ML models.

**Figure 5 animals-13-01596-f005:**
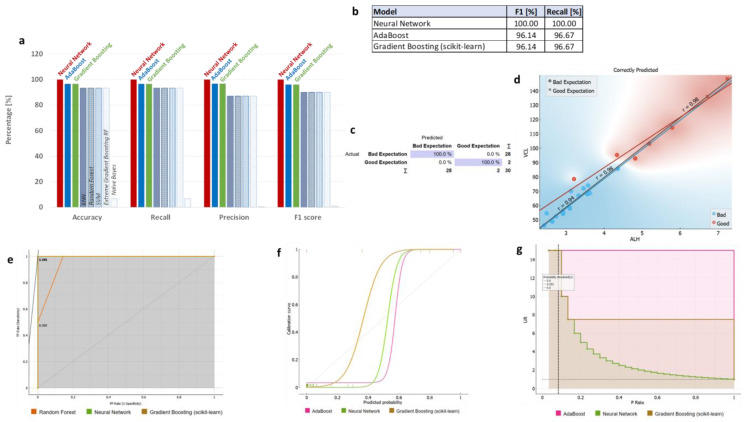
Classification power of the learning algorithms used to classify the sperm dataset based on classifying label “Suggested Good Quality”. Predictions were made on a dataset in a 5-fold cross-validation process, where the total dataset was split into five folds, and each was once used as a test set, while the remaining ones were used as a validation set. Each cross-validation repetition used a different seed. Metrics presented are Orange cross-validation algorithm performance parameters: (**a**) The percentage accuracy, recall, precision, and F1-score of the different ML algorithm models are compared with each other; (**b**) F1-score and recall from the Neural Network (Multi-layer Perceptron scikit-learn algorithm, MLP-NN), Ad Boost (Orange), and Gradient Boosting (scikit-learn) learning models considerably outperformed the kNN, Random Forest, SVM, Extreme Gradient Boosting Random Forest (RF), and Naïve Bayes algorithms; (**c**) Confusion matrix of the Neural Network (MLP) model showing actual probabilities of correct and incorrect classifications for the progressive-motility-based “Suggested Good Quality” class, where blue shading shows correctly classified predictions; (**d**) Example correlation plot of parameters selected based on the best classifier, with the strongest correlation (*p* < 0.001 for r and r_Good,_ r_Bad_); (**e**) Receiver operating characteristic (ROC) curves; (**f**) Calibration curves; (**g**) Performance curve of the compared ML models.

**Figure 6 animals-13-01596-f006:**
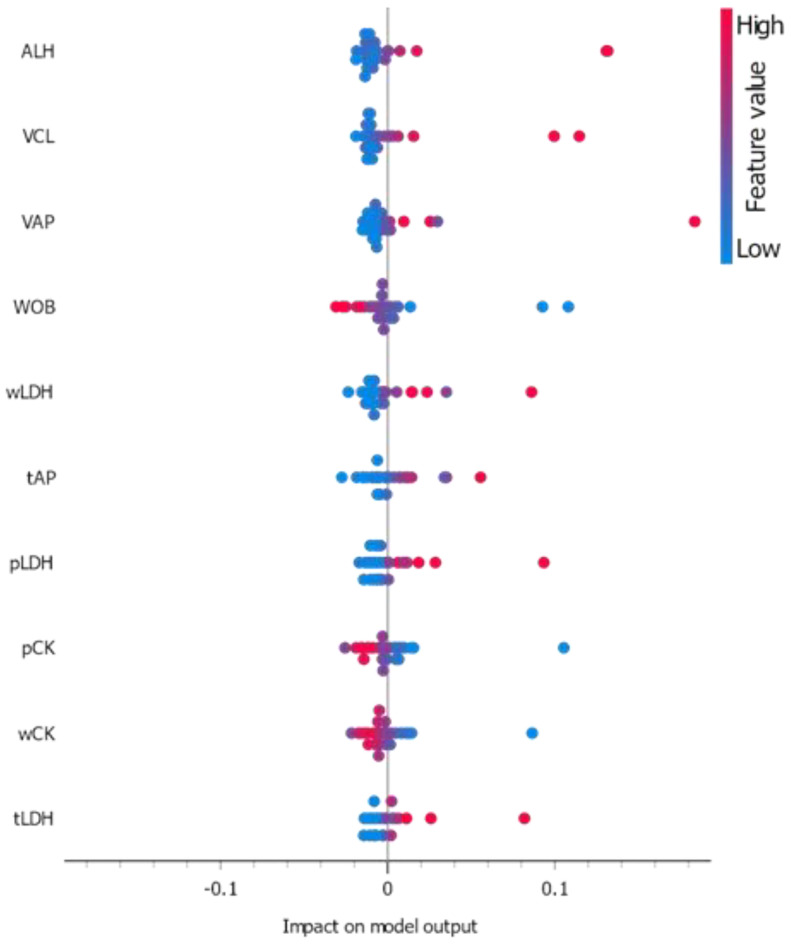
Classification Model explanation plot. The plot shows the features that contribute the most to the prediction for the specific class “Good expectations” (classification feature “Suggested Good Quality”) and how their values affect this prediction. Red colour dots are representative of the higher feature value, while blue ones are for lower values. Based on their position towards 0 on the *x*−axis, positive feature values influence the prediction of the specified class, whilst negative values influence classification against the selected class.

**Table 1 animals-13-01596-t001:** Sperm parameters of diluted semen stored at 4 °C for 6 h.

Traits	Parameter “Quality”: “Good”	Parameter “Quality”: “Bad”	Significant Differences	*p*-Value
Motility				
Total motility, %	99.23 ± 0.34	75.32 ± 2.71	0.000	*p* < 0.001
Progressive motility, %	69.42 ± 6.05	25.68 ± 2.32	0.000	*p* < 0.001
Non-progressive motility, %	29.81 ± 5.81	49.64 ± 1.06	0.000	*p* < 0.001
Static spz, %	0.77 ± 0.34	24.68 ± 2.71	0.000	*p* < 0.001
Velocity motion parameters				
Rapid spz,	56.57 ± 8.91	13.41 ± 1.44	0.000	*p* < 0.001
Medium spz,	31.94 ± 5.38	26.50 ± 1.98	0.248	n.s.
Slow spz,	10.72 ± 3.57	35.32 ± 1.71	0.000	*p* < 0.001
Velocity parameters				
VCL, µm/s	109.75 ± 9.43	60.78 ± 2.03	0.000	*p* < 0.001
VAP, µm/s	46.35 ± 6.65	24.67 ± 1.65	0.000	*p* < 0.001
VSL, µm/s	52.86 ± 4.85	34.00 ± 1.67	0.000	*p* < 0.001
STR, %	44.18 ± 6.64	41.64 ± 1.84	0.606	n.s.
LIN, %	47.44 ± 4.00	55.22 ± 1.73	0.051	n.s.
WOB, %	59.59 ± 2.84	59.76 ± 0.83	0.936	n.s.
ALH, µm	5.33 ± 0.52	3.14 ± 0.09	0.000	*p* < 0.001
BCF, Hz	5,43 ± 0.32	4.68 ± 0.17	0.041	*p* < 0.05
Morphological characteristics				
Live normal spz, %	85.86 ± 2.29	80.14 ± 1.23	0.033	*p* < 0.05
Head defects spz, %	4.14 ± 1.03	5,68 ± 0.79	0.333	n.s.
Midpiece defects spz, %	5.88 ± 1.58	9.65 ± 0.84	0.040	*p* < 0.05
Tail defects spz, %	4.14 ± 1.32	4.37 ± 0.58	0.862	n.s.

n.s.—no significant difference.

**Table 2 animals-13-01596-t002:** Enzyme activities (UI/l) in seminal plasma (sp), water (w), and Triton X100 sperm extracts (t).

Factor		AP (Alkaline Phosphatase)	LDH (Lactate Dehydrogenase)	CK (Creatine Kinase)	GGT (Gamma-Glutamyl Transferase)
Semen “Quality” Parameter	Semen Fractions				
Good	seminal plasma (sp)	16.00 ± 3.49 ^b^	271.29 ± 81.23	65.29 ± 13.35 ^a^	385.29 ± 56.29 ^ab^
	water sperm extract (w)	36.86 ± 8.84 ^ab^	241.70 ± 74.35	49.57 ± 11.79 ^ab^	308.00 ± 53.90 ^b^
	Triton sperm extract (t)	19.29 ± 2.64 ^b^	132.86 ± 32.51	17.29 ± 4.19 ^b^	693.00 ± 49.34 ^a^
Bad	seminal plasma (sp)	89.61 ± 20.75 ^ab^	244.17 ± 53.39	77.44 ± 7.24 ^a^	286.16 ± 64.35 ^b^
	water sperm extract (w)	59.44 ± 6.94 ^ab^	179.60 ± 43.99	77.45 ± 5.41 ^a^	227.61 ± 55.51 ^b^
	Triton sperm extract (t)	26.74 ± 3.25 ^ab^	85.99 ± 21.22	63.57 ± 8.19 ^a^	285.44 ± 65.78 ^b^
mean ± SEM		50.53 ± 6.30	180.50 ± 21.21	66.10 ± 3.77	312.07 ± 30.51
Semen “Quality” (Bad × Good)		0.020 (*p* < 0.05)	0.369 (n.s.)	0.001 (*p* < 0.01)	0.06 (n.s.)
Semen fractions (sp × w × t)		0.07 (n.s.)	0.010 (*p* < 0.05)	0.039 (*p* < 0.05)	0.200 (n.s.)
Semen “Quality” × Semen fractions		0.001 (*p* < 0.01)	0.075 (n.s.)	0.001 (*p* < 0.01)	0.008 (*p* < 0.05)

Note: Values with different superscripts in the corresponding column (biochemical parameter) are significantly different (*p* < 0.05), while ones with the same superscripts are not different (*p* > 0.05).

## Data Availability

The datasets used and analysed during the current study is available from the corresponding author on reasonable request.

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
