# Peer review of "Machine Learning Approach for Muscovy Duck (Cairina moschata) Semen Quality Assessment"

_animals, 2023, doi:10.3390/ani13101596_

Round 1

Reviewer 1 Report

Interesting study that promotes the potential of using machinel earning and neural networks to evaluate sperm samples for fertiliztion potential. 

The simple summary must fit the purpose of a "simple summary," which I don't believe is the case. In the abstract, as well as throughout the paper, most acronyms are not defined. I suggest creating a table with all abbreviations provided as they are numerous in this paper (with most never being defined). There are multiple grammatical errors, mostly concerning syntax.  Also, the figures are too small and the quality could be improved as the clarity is not great upon zooming. I also believe that the the machine learning aspect and validation proceedures may need to be simplified for readers of Animals as they tend to be more biologically focused and likely unfamiliar with the terminology or use of computer algorthtms/programs. 

Some considerations by line: 

79: somatic methilation is similar to the "animal one" should be mammalian (birds are animals afterall). 

141: insert "of the" between consisted and following (this is just one of several syntax errors in the paper). 

143: 0.4 glycerol ? %, mL, ? 

165: "Freeze" should be replaced with "Stored at -20"

494: "life stock" to livestock

506: reword, nonsensical 

520: reword, syntax

523: unlike to "unlikely" 

Author Response

Comments and Suggestions for Authors

Interesting study that promotes the potential of using machine learning and neural networks to evaluate sperm samples for fertilization potential.

The simple summary must fit the purpose of a "simple summary," which I don't believe is the case. In the abstract, as well as throughout the paper, most acronyms are not defined. I suggest creating a table with all abbreviations provided as they are numerous in this paper (with most never being defined). There are multiple grammatical errors, mostly concerning syntax.  Also, the figures are too small and the quality could be improved as the clarity is not great upon zooming. I also believe that the machine learning aspect and validation procedures may need to be simplified for readers of Animals as they tend to be more biologically focused and likely unfamiliar with the terminology or use of computer algorithms/programs.

Authors’ reply in italic:

We appreciate very much all suggestions and remarks made on the manuscript as well as the overall evaluation of manuscript contribution to the field!

  • The Simple Summary was rewritten.
  • Acronyms were inserted throughout the text body in order of first appearance, and a dedicated Acronyms table was added at the end of the text as suggested.
  • Two figures containing multiple results had very small numbers that were readily seen in the Word version of the manuscript, but not in the PDF exports regardless of the resolution selected and were therefore corrected to contain higher sized fonts. We added the original TIF/PDF files for clarity.
  • The font size was also enlarged in all Tables.

79: somatic methilation is similar to the "animal one" should be mammalian (birds are animals afterall). - corrected

141: insert "of the" between consisted and following (this is just one of several syntax errors in the paper). - accept

143: 0.4 glycerol ? %, mL, ? - corrected

165: "Freeze" should be replaced with "Stored at -20" - accept

494: "life stock" to livestock - corrected

506: reword, nonsensical - corrected

520: reword, syntax - corrected

523: unlike to "unlikely" - corrected

All of the above were corrected! The long paragraphs were rewritten for clarity, the grammar and syntax were corrected throughout the manuscript.

The ML methodology was extended in the Material and Methods, as well as in the Results and Discussion sections.

Reviewer 2 Report

Machine learning approach for Muscovy duck (Cairina moschata) semen quality assessment

Comments

The manuscript requires English-language proofing. It has a lot of grammatical mistakes as well as long and confusing sentences.

The limitations of manual screening of Muscovy duck semen samples are not convincing enough to conduct an elaborate study to develop ML models. From an economic standpoint, the feasibility of  ML models over manual screening for Muscovy duck production is not explained.

Title

Cairina moschata should be in italics.

Abstract

Multiple extremely long and confusing sentences are used. Please edit it appropriately. Ex: Lines 18-23: It is an extremely long and confusing sentence.

Multiple abbreviations without expanding are used. In abstract, use only the most important abbreviations and their expansions. Ex: Line 25: All the abbreviations listed need to be expanded.

Introduction

Throughout the manuscript ‘commas’ are used inappropriately and is affecting the flow and meaning of the sentence. Please correct it.

Expand all abbreviations when they appear in the text for the first time. Ex: Line 41: Expand CASA.

Line 44: What do you mean by “offspring bird rates and sizes”? Please be clear.

Line 47: Use abbreviation ML here instead of line 48.

Lines 48-50: Confusing sentence. Correct the grammatical errors.

Line 50: Remove ‘learning’ after ML.

Line 57: Change as ‘frozen semen’ instead of ‘’conserved sperm’.

Lines 61-64: The sentence is confusing. Edit it appropriately.

Lines 78-80: Correct the grammatical errors.

Lines 83-85: The first part, do you specifically mean in birds. If so, please mention birds there.

Line 90: Do you mean the extend or length of semen storage? If so, change it. Also, use ‘type of extender’ instead of ‘kind of extender’.

Lines 98-99: It’s not clear what do you mean? Loss of sperm head??

Lines 86-102: Are the studies cited in this paragraph directly or indirectly showing the effect of DNA methylation on the changes in sperm morphology reported? What the authors meant is not conveyed properly.

Lines 104-108: It’s not clear why these non-morphological parameters are mentioned here.

Line 109: The limitations of manual screening of semen samples are not convincing enough to conduct an elaborate study to develop ML models. From an economic standpoint, is the ML models feasible over manual screening for Muscovy duck production?

Results

Lines 273-274: It’s repeated twice. Correct it.

Table 2: Enzyme name expansion needs to be included.

Line 86: Correct it as ‘while ones with the same superscripts are not different (p > 0.05).’

Figure 1: Are the correlations represented significant? P values are not shown.

Author Response

Authors’ reply in italic:

We appreciate very much all suggestions and remarks made on the manuscript as well as the overall evaluation of manuscript contribution to the field!

All of the raised issues with grammar and syntax were corrected! The long paragraphs were rewritten for clarity, the grammar and syntax were corrected throughout the manuscript.

The ML methodology was extended in the Material and Methods, as well as in the Results and Discussion sections.

As for the “The limitations of manual screening of Muscovy duck semen samples are not convincing enough to conduct an elaborate study to develop ML models. From an economic standpoint, the feasibility of  ML models over manual screening for Muscovy duck production is not explained.”, we have explained in the Introduction, MM and Discussion sections that the possibility for one to train a large dataset could be later used by others providing only limited part of the parameters to the software and still obtaining predictions with reasonably good value, which is the ultimate goal to every ML approach. The other benefit of the study is the introduction of DNA methylation as additional parameter that could potentially help in cases with good morphology.

Line 44: What do you mean by “offspring bird rates and sizes”? - corrected

Line 47: Use abbreviation ML here instead of line 48. - accept

Lines 48-50: Confusing sentence. Correct the grammatical errors. - corrected

Line 50: Remove ‘learning’ after ML. - corrected

Line 57: Change as ‘frozen semen’ instead of ‘’conserved sperm’. - corrected

Lines 61-64: The sentence is confusing. Edit it appropriately. - changed

Lines 78-80: Correct the grammatical errors.  -corrected

Lines 83-85: The first part, do you specifically mean in birds. If so, please mention birds there. - corrected

Line 90: Do you mean the extend or length of semen storage? If so, change it. Also, use ‘type of extender’ instead of ‘kind of extender’. - corrected

Lines 98-99: It’s not clear what do you mean? Loss of sperm head?? - corrected

Lines 86-102: Are the studies cited in this paragraph directly or indirectly showing the effect of DNA methylation on the changes in sperm morphology reported? What the authors meant is not conveyed properly. - changed

Lines 104-108: It’s not clear why these non-morphological parameters are mentioned here. - corrected

Line 109: The limitations of manual screening of semen samples are not convincing enough to conduct an elaborate study to develop ML models. From an economic standpoint, is the ML models feasible over manual screening for Muscovy duck production? - changed

Results

Lines 273-274: It’s repeated twice. Correct it. - corrected now

Table 2: Enzyme name expansion needs to be included. - corrected

Line 86: Correct it as ‘while ones with the same superscripts are not different (p > 0.05).’ - corrected now

Figure 1: Are the correlations represented significant? P values are not shown. - accept

Reviewer 3 Report

That is an interesting manuscript related to duck semen analysis. It is in general well written and clear, but there are some points that needs to be addressed before acceptance.

1. Simple Summary - Reading this section is quite difficult. I suggest that it be completely rewritten.

2. Abstract - Although the content is well written, I missed numerical values for both the analysis results and the correlations found. I believe that the demonstration of such values would be welcome to illustrate the main findings and conclusions.

3. Introduction - From my point of view, the introduction is very well written, if a little long. I have, however, a criticism to make, since it presents numerous considerations for the analyzes conducted by CASA and for the evaluation of DNA methylation. However, the introduction does not mention the enzymes evaluated in the work through biochemical analysis. My suggestion would be to briefly summarize the text regarding DNA methylation, and include considerations that justify the biochemical analysis.

4. Methods:

- The ethical statement should come at the beginning of the methodology, before the Birds and Rearing topic.

- At the Birds and Rearing topic, please include geographical coordinates where the experiments were conducted. This is an important information because you are researching on a seasonal species.

- Has the semen extender formulation used previously been used for ducks or other birds? If so, please mention it in the text and reference it.

- In sperm analysis, please provide CASA settings used for duck sperm analysis.

- I miss references for enzyme assays. 

- Also, provide references for DNA methylation analysis.

5. Results:

- In Table 1, I suggest that the authors indicate in the first line the number of animals (or ejaculates?) classified as Bad or Good.

- In texts referring to correlations between variables (eg line 290), please indicate the value of the correlation coefficient (r). This indication is important to illustrate the weak, moderate or strong correlations found.

6. Discussion:

- Authors should avoid such long paragraphs, as this makes them difficult to read. For example, the first paragraph of the discussion should be divided according to the topic addressed.

- It is necessary to emphasize that this first long paragraph of the discussion quotes sentences that were already addressed in the introduction of the work. Therefore, it is important that the authors are not repetitive and choose to use such sentences in one session or another.

- In the discussions and conclusions, I believe that the authors should mention the predictive inefficiency of enzyme analysis, given that this was one of the objectives proposed by the work.

Author Response

 Authors’ reply in italic:

We appreciate very much all suggestions and remarks made on the manuscript as well as the overall evaluation of manuscript contribution to the field!

All of the raised issues with grammar and syntax were corrected! The long paragraphs were rewritten for clarity, the grammar and syntax were corrected throughout the manuscript. The enzyme section was also increased as requested. CASA settings of the software and more detail on the protocol were provided as well.

The ML methodology was extended in the Material and Methods, as well as in the Results and Discussion sections.

v

1. Simple Summaryrewritten.

2. Abstract - written

3. Introduction - accept and chaned

4. Methods:

- The ethical statement should come at the beginning of the methodology, before the Birds and Rearing topic. - accept

- At the Birds and Rearing topic, please include geographical coordinates where the experiments were conducted. ..... - included in methods

- Has the semen extender formulation used previously been used for ducks or other birds? If so, please mention it in the text and reference it. - changed

- In sperm analysis, please provide CASA settings used for duck sperm analysis. - accept

- I miss references for enzyme assays. - accept

- Also, provide references for DNA methylation analysis. - provided

5. Results:

- In Table 1, ..... - corrected

- In texts referring to correlations between variables ....... - corrected

6. Discussion:

- Authors should avoid such long paragraphs, as this makes them difficult to read. ..... - corrected

- It is necessary to emphasize that this first long paragraph of the discussion quotes sentences .... - corrected

- In the discussions and conclusions, .... - corrected

Reviewer 4 Report

Dear author,

Bellow you will find comments  that you have to follow and to do in  your manuscript

P1L21-22:  [alkaline phosphatase (ALP), creatine kinase (CK), lactate dehydrogenase (LDH) and γ-glutamyl-21 transferase (GGT)], instead of (alkaline phosphatase (ALP), creatine kinase (CK), lactate dehydrogenase (LDH) and γ-glutamyl-21 transferase (GGT)),

P2L94: Gerzilov et al. (2021) reported instead of Gerzilov et al. reported

P2L94: HIA-1 extenders (22) instead of HIA-1 extenders

P4L148-149: Refer the conditions (temperature, etc) that semen samples were transported to the Institute of Biology and Immunology of Reproduction at Bulgarian Academy of Sciences, Sofia

P4L156-158: In each ejaculate 100 sperm cells were analyzed (magnification x..........) under Olympus BX50 microscope (Nikon, Tokyo, Ja-156 pan) instead of The measurements were performed on 100 randomly selected sperm in each ejaculate. Determine the magnification you were analyzed the ejaculates

P4L171-172: were used to analyze the activity of enzymes instead of were used for analysed on the activity of enzymes

P4L185-193 : The whole procedure of assessment of DNA methylation has to be rewritten more clear so to be more understandable.

Machine learning sperm parameters classification and Performance Assessment has to  go before  Statistical analysis

P9L273-276: It was found that the LDH and CK enzyme activities were significantly different 273 in the sperm media (p<0.05 and p<0.05, respectively). The sentence is written twice.

INTRODUCTION : is not clear the aim of the work. Also you have to emphasize more the way that the Machine learning sperm parameters classification will be used in farm practice

MATERIALS & METHODS:

·  In the parameters that an ejaculate has to fulfill  there is no report on the concentration of the ejaculate As far as I know the CASA gives report also for the concentration.

·  The sperm cells you refer that were evaluated with an Olympus BX50 microscope but you are not referring the magnification you have used.

·  Machine learning sperm parameters classification is not clear the evaluation framework and how many steps involved in the proposed framework

·  Performance Assessment To verify the accuracy of the models generated, how the performance of each model was evaluated is not clear

RESULTS: Is not clear the model performance in predicting semen quality

Author Response

 Authors’ reply in italic:

We appreciate very much all suggestions and remarks made on the manuscript as well as the overall evaluation of manuscript contribution to the field!

All of the raised issues with grammar and syntax were corrected! The long paragraphs were rewritten for clarity, the grammar and syntax were corrected throughout the manuscript.

The ML methodology was extended in the Material and Methods, as well as in the Results and Discussion sections.

v

P1L21-22: corrected

P2L94: corrected

P2L94: HIA-1 extenders (22) instead of HIA-1 extenders - corrected

P4L148-149: corrected

P4L156-158: accepted and change

P4L171-172: were used to analyze the activity of enzymes instead of were used for analysed on the activity of enzymes - corrected

P4L185-193 : - rewrotten

P9L273-276:  corected

INTRODUCTION : was corrected

MATERIALS & METHODS:

  • In the parameters that an ejaculate has to fulfill there is no report on the concentration of the ejaculate As far as I know the CASA gives report ... - accepted
  • The sperm cells you refer that were evaluated with an Olympus BX50 microscope but you are not referring the magnification you have used. - changed
  • Machine learning sperm parameters classification is not clear the evaluation framework and how many steps involved in the proposed framework - corrected
  • Performance Assessment To verify the accuracy of the models generated, how the performance of each model was evaluated is not clear - corrected

RESULTS: rewrotten

Round 2

Reviewer 2 Report

The authors have addressed all comments given by this reviewer and have made the necessary changes in the manuscript. This reviewer is convinced by the responses.   

Reviewer 4 Report

Dear author 

Reading the corrected version of your manuscript I noticed that you have accepted  my comments by which I tried to help you.